# Induction of Labor in Twins—Double Trouble?

**DOI:** 10.3390/jcm12052041

**Published:** 2023-03-04

**Authors:** Miriam Lopian, Lior Kashani-Ligumsky, Ronnie Cohen, Izaak Wiener, Bat-Chen Amir, Yael Gold Zamir, Ariel Many, Hadar Rosen

**Affiliations:** 1Department of Obstetrics and Gynecology, Mayanei Hayeshua Medical Center, Bnei Brak 51544, Israel; 2Sackler School of Medicine, Tel Aviv University, Tel Aviv 69978, Israel

**Keywords:** induction of labor, twin pregnancies, cesarean delivery, prostaglandin E1

## Abstract

Objective: To determine and compare the safety and efficacy of different methods of induction of labor in twin gestations and their effect on maternal and neonatal outcomes. Methods: A retrospective observational cohort study was conducted at a single university-affiliated medical center. Patients with a twin gestation undergoing induction of labor at >32 + 0 weeks comprised the study group. Outcomes were compared to patients with a twin gestation at >32 + 0 weeks who went into labor spontaneously. The primary outcome was cesarean delivery. Secondary outcomes included operative vaginal delivery, postpartum hemorrhage, uterine rupture, 5 min APGAR < 7, and umbilical artery pH < 7.1. A subgroup analysis comparing outcomes for the induction of labor with oral prostaglandin E1 (PGE1), IV Oxytocin ± artificial rupture of membranes (AROM), and extra-amniotic balloon (EAB)+ IV Oxytocin was performed. Data were analyzed using Fisher’s exact test, ANOVA, and chi-square tests. Results: 268 patients who underwent induction of labor with a twin gestation comprised the study group. 450 patients with a twin gestation who went into labor spontaneously comprised the control group. There were no clinically significant differences between the groups for maternal age, gestational age, neonatal birthweight, birthweight discordancy, and non-vertex second twin. There were significantly more nulliparas in the study group compared to the control group (23.9% vs. 13.8% *p* < 0.001). The study group was significantly more likely to undergo a cesarean delivery of at least one twin (12.3% vs. 7.5% OR, 1.7 95% CI 1.04–2.85 *p* = 0.03). However, there was no significant difference in the rate of operative vaginal delivery (15.3% vs. 19.6% OR, 0.74, 95% CI 0.5–1.1 *p* = 0.16), PPH (5.2% vs. 6.9% OR, 0.75 95% CI 0.39–1.42 *p* = 0.37), 5-min APGAR scores < 7 (0% vs. 0.2% OR, 0.99 95%CI 0.99–1.00 *p* = 0.27), umbilical artery pH < 7.1 (1.5% vs. 1.3% OR, 1.12 95% CI 0.3–4.0), or combined adverse outcome (7.8% vs. 8.7% OR, 0.93 95% CI 0.6–1.4 *p* = 0.85). Furthermore, there were no significant differences in the rates of cesarean delivery or combined adverse outcomes in patients undergoing induction with oral PGE1 compared to IV Oxytocin ± AROM (13.3% vs. 12.5% OR, 1.1 95% CI 0.4–2.0 *p* = 1.0) (7% vs. 9.3% OR, 0.77 95% CI 0.5–3.5 *p* = 0.63 ) or EAB+ IV Oxytocin (13.3% vs. 6.9% OR, 2.1 95% CI 0.1–2.1 *p* = 0.53) (7% vs. 6.9% OR, 1.4 95% CI 0.15–3.5 *p* = 0.5) or between patients undergoing induction of labor with IV Oxytocin ± AROM and EAB+ IV Oxytocin (12.5% vs. 6.9% OR, 2.1 95% CI 0.1–2.4 *p* = 0.52) (9.3% vs. 6.9% OR, 0.98 95% CI 0.2–4.7 *p* = 0.54). There were no cases of uterine rupture in our study. Conclusions: Induction of labor in twin gestations is associated with a two-fold increased risk of cesarean delivery, although this is not associated with adverse maternal or neonatal outcomes. Furthermore, the method of induction of labor used does not affect the chances of success nor the rate of adverse maternal or neonatal outcomes.

## 1. Introduction

The prevalence of twin pregnancies has increased over the past several decades, largely due to the development and widespread availability of Artificial Reproductive Technology and advanced maternal age [1]. Indeed, in 2020, twin births accounted for over 3% of live births compared to 1.9% in 1980 [2]. Twin pregnancies are at an increased risk of a range of obstetric complications, including hypertensive disorders of pregnancy [3,4], gestational diabetes [5], growth restriction [6,7], and intrauterine fetal demise [8]. These conditions are often indications for the delivery and induction of labor. Furthermore, although twin pregnancies are at an increased risk of spontaneous preterm birth [9], evidence of increased perinatal morbidity and mortality near term [10] has led the American College of Obstetricians and Gynecologists to recommend elective delivery of uncomplicated dichorionic twins between 38 + 0–38 + 6 weeks and monochorionic twins between 34 + 0–37 + 6 weeks [11]. Therefore, another common indication for induction of labor in twin pregnancies is gestational age. Whilst there is much data in the literature regarding the safety and efficacy of various methods of induction of labor in singleton gestations, data in twin gestations is sparse, and as such, the management of induction of labor in twin gestations is largely derived from data derived from singleton gestations. That said, there are distinct differences between a singleton and twin trial of labor. Rates of spontaneous vaginal delivery are lower in twin gestations [12,13], and recent evidence suggests that the rate of labor progress both in the first and second stages of labor differs in twin gestations [14,15]. Furthermore, there are theoretical concerns that an overdistended uterus in twin gestations might be more prone to uterine rupture with induction of labor compared to singleton gestations [16]. 

The aim of this study is to determine and compare the safety and efficacy of different methods of induction of labor in twin gestations and their effect on maternal and neonatal outcomes. 

## 2. Materials and Methods

A retrospective observational cohort study was conducted at a single university-affiliated medical center from 2012–2022. This center has a delivery ward with approximately 11,000 deliveries per year, with a twin delivery rate of 1.2%. 

Pre-natal and post-natal outcomes were collected from a computerized database for patients carrying twin gestations who underwent a spontaneous or induced trial of labor during the study period. The study group consisted of patients with a twin gestation with a gestational age greater than 32 + 0 weeks of gestation who underwent a medical induction of labor with either oral prostaglandin E1 (Cytotec), IV Oxytocin±artificial rupture of membranes (AROM), or mechanical induction with an extra-amniotic balloon + IV Oxytocin. Maternal and neonatal outcomes were compared to those of twin gestations who went into labor spontaneously. 

The primary outcome was the cesarean delivery of either one or both twins. Secondary outcomes included mode of delivery (spontaneous delivery or vacuum extraction) and combined adverse outcome (postpartum hemorrhage (PPH), uterine rupture, umbilical artery pH < 7.1, and APGAR < 7 at 5 min for either twin). 

Further analyses were conducted to compare maternal and neonatal outcomes in twin gestations according to the method of induction used. Data was collected regarding indications for cesarean delivery in patients undergoing induction of labor. 

Patients with a previous cesarean delivery or contraindications to vaginal delivery were excluded from the study. Contraindications to vaginal delivery of twins in our center include any contraindication to vaginal delivery, presenting twin non-vertex, a sonographic estimated fetal weight of either twin less than 1500 g, gestational age less than 32 + 0 weeks, and twin discordancy of greater than 20% in favor of the second twin in a non-vertex presentation. 

Patients with a poor Bishop score (<6) on admission were induced with either oral PGE1 or an extra-amniotic balloon + IV Oxytocin. Patients with a Bishop score ≥6 were induced with IV Oxytocin ± AROM. The protocol for induction of labor with PGE1 at our center involves the administration of an initial dose of 50 micrograms of PGE1 orally. This dose is repeated every four hours until active labor develops, up to a maximal dose of 300 micrograms. Contraindications to the use of PGE1 in our center are the same in singleton and twin gestations and include the presence of any uterine scar, grandmultiparity (parity > 5), or non-reassuring fetal status. The IV Oxytocin induction protocol at our center involves the administration of 2.5 mu/min of IV Oxytocin which is increased at increments of 2.5 mu/min every 20 min until a maximum of 22.5 mu/min. AROM is performed at the attending physician’s discretion. Induction of labor with an extra-amniotic balloon involves the passage of a 22-Gauge Foley catheter through the internal os of the cervix. The catheter balloon is filled with 60 mL of normal saline and remains in place for a maximum of 24 h.

Statistical analysis was performed using IBM SPSS Statistics for Windows, Version 26.0. Armonk, NY, USA: IBM Corp. Continuous variables were analyzed using the independent samples *t*-test. Non-continuous variables were analyzed using Fishers’ exact test, ANOVA, and Chi-square test. The local ethical review board approved the study. Approval number MHMC-0025-21.

## 3. Results

268 patients comprised the study group of twin gestations undergoing induction of labor. Of those, 142 (53.3%) underwent induction of labor with prostaglandin E1(Cytotec) according to the local hospital protocol, 96(35.8%) patients were induced with IV Oxytocin ± AROM, and 29 (10.8%) patients received an extra-amniotic balloon + IV Oxytocin. Maternal and neonatal outcomes were compared to 450 patients with a twin gestation who went into labor spontaneously. There were no clinically significant differences between the groups in mean maternal age (29.9 ± 5.7 vs. 30.7 ± 5.6 *p* = 0.03), multiparity (56% vs. 58% *p* = 0.59), gestational age at delivery (36.9 ± 1.5 vs. 36. ± 1.5 *p* = 0.8), birthweight (2562 ± 402 vs. 2565 ± 390 *p*= 0.94), birthweight discordancy >20% (12.7% vs. 9.1% *p* = 0.13), and the rate of non-vertex second twins (42.4% vs. 44.2% *p* = 0.64) (Table 1). There was a significantly higher prevalence of nulliparas (23.9% vs.13.8% *p* < 0.001) and a lower prevalence of grand multiparas (20.1% vs. 28.2% *p* = 0.01) in the group undergoing induction of labor. 

Patients undergoing induction of labor with a twin gestation were significantly more likely to undergo a cesarean delivery of at least one twin (12.3% vs. 7.5% OR, 1.7 95% CI 1.04–2.85 *p* = 0.03). There was no significant difference in the rate of spontaneous vaginal delivery of both twins (72.3% vs. 72.7% OR, 0.99 95% CI 0.7–1.4 *p* = 0.93) or operative vaginal delivery of at least one twin (15.3% vs. 19.6% OR, 0.74, 95% CI 0.5–1.1 *p* = 0.16). There were no differences between the groups in the rate of PPH (5.2% vs. 6.9% OR, 0.75 95% CI 0.39–1.42 *p* = 0.37), and there were no cases of uterine rupture in this study population. 

With regards to neonatal outcomes, there were no significant group differences in 5-min APGAR score < 7 (0% vs. 0.2% OR, 0.99 95%CI 0.99–1.00 *p* = 0.27) or umbilical artery pH < 7.1 (1.5% vs. 1.3% OR, 1.12 95% CI 0.3–4.0). Overall, there were also no differences in the rate of combined adverse maternal and neonatal outcomes (7.8% vs. 8.7% OR, 0.93 95% CI 0.6–1.4 *p* = 0.85) (Table 2) 

A subgroup analysis was performed comparing the different methods of labor induction used, oral PGE1 (Cytotec), IV Oxytocin ± AROM, or extra-amniotic balloon + IV Oxytocin. There were no significant between-group differences in mean maternal age, parity, gestational age at delivery, birthweight, birthweight discordancy >20%, or the rate of non-vertex second twins in patients undergoing induction of labor (Table 3). There were no significant differences in the rates of cesarean delivery in patients undergoing induction with PGE1 compared to Oxytocin ± AROM (13.3% vs. 12.5% OR, 1.1 95% CI 0.4–2.0 *p* = 1.0), extra-amniotic balloon + IV Oxytocin (13.3% vs. 6.9% OR, 2.1 95% CI 0.1–2.1 *p* = 0.53), or between patients undergoing induction of labor with Oxytocin ±AROM and extra-amniotic balloon + IV Oxytocin (12.5% vs. 6.9% OR, 2.1 95% CI 0.1–2.4 *p* = 0.52). 

There were also no significant differences in the rates of combined adverse outcomes in patients undergoing induction with PGE1 compared to Oxytocin ± AROM (7% vs. 9.3% OR, 0.77 95% CI 0.5–3.5 *p* = 0.63), extra-amniotic balloon + IV Oxytocin (7% vs. 6.9% OR, 1.4 95% CI 0.15–3.5 *p* = 0.5), or between patients undergoing induction of labor with IV Oxytocin ± AROM and extra-amniotic balloon + IV Oxytocin (9.3% vs. 6.9% OR, 0.98 95% CI 0.2–4.7 *p* = 0.54) (Table 4). There were no cases of uterine rupture in twin gestations undergoing induction of labor. Indications for cesarean delivery in patients undergoing induction of labor are listed in Table 5. Cesarean deliveries that were performed for complications of the second twin include four cases of umbilical cord prolapse of the second twin, three cases of placental abruption of the second twin, one case of hand presentation of the second twin, and one case of non-reassuring fetal heart rate tracing of the second twin. 

## 4. Discussion

Our findings indicate that patients with twin pregnancies who undergo induction of labor after 32 + 0 weeks are twice as likely to undergo cesarean delivery than those who enter labor spontaneously. However, they are not at increased risk of experiencing adverse maternal outcomes, including unplanned operative vaginal delivery, uterine rupture, and PPH, nor are they at increased risk of having adverse neonatal outcomes, including umbilical artery pH < 7.1 and APGAR scores of <7 at 5 min. Furthermore, despite the increased risk of cesarean delivery in this population, overall, they have a good chance of achieving vaginal delivery (88.2%). 

Despite the prevalence of this clinical scenario, little data exists regarding maternal and neonatal outcomes following the induction of labor in twin pregnancies. Results from larger studies that have been conducted regarding the safety and feasibility of induction of labor in twin gestations are summarized in Table 6 and include the data from this study [17,18,19,20,21,22]. 

The published success rate for induction of labor in twin pregnancies ranges between 59.5% and 81.0% [17,18,19,20,21,22]. We report a success rate of 87.7%, the highest reported in the scientific literature thus far. Several factors may account for our higher success rates. One is due to the characteristics of our study population, which has a young mean maternal age of 30.2 years old and low rates of nulliparity (17.5%). Indeed, advanced maternal age and nulliparity are known risk factors for a failed trial of labor in twin gestations [23]. Furthermore, due to a cultural desire for higher parity, our patient population is highly motivated for vaginal delivery. Over 90% of patients with twin gestations who were eligible to undergo a trial of labor in our center chose to do so, and our patients’ desire to avoid cesarean delivery may be responsible for physician bias when managing labor, e.g., opting to perform an internal podalic version or total breech extraction rather than cesarean delivery for a non-vertex second twin. 

There is much debate in the literature concerning the impact of induction of labor in singleton gestations on the risk of cesarean delivery. Some studies report that induction of labor is associated with an increased risk of cesarean delivery compared to the spontaneous onset of labor [24], whilst others have found that induction of labor may reduce the risk of cesarean delivery compared to expectant management, even when performed for non-medical indications [25].

The few studies on the induction of labor in twins have traditionally used singleton pregnancies undergoing induction of labor or twin pregnancies in spontaneous labor as the control groups [17,18,19,20,21,22] (Table 6). Most studies concur with the results of this present study and demonstrate a higher risk of cesarean delivery in twin gestations undergoing induction of labor compared to both these control groups [17,18,19,20,21]. Similar to this study, no studies reported an increased risk of maternal or neonatal adverse outcomes or any cases of uterine rupture in twins undergoing induction of labor [17,18,19,20,21,22]. 

Possible reasons that have been given for the increased rate of cesarean delivery in twins undergoing induction of labor include uterine inertia to uterotonic medications as a result of overdistention [26] and confounding risk factors for cesarean delivery in patients undergoing induced rather than spontaneous labor, e.g., comorbidities and non-reassuring fetal status. In our study, the higher prevalence of nulliparas (23.9%) in the group undergoing induction of labor compared to spontaneous labor (13%) certainly could contribute to this difference, as nulliparity is a known risk factor for cesarean delivery [23]. 

We selected patients with twin pregnancies undergoing spontaneous labor as our control group due to the inherent distinct differences between twin and singleton gestations in terms of maternal and neonatal outcomes [27], rates of vaginal delivery [12,13], normal progress of labor [14,15], and complications owing from the delivery of a second twin [27]. Furthermore, data from this comparison may be useful for counseling patients with twin pregnancies who are presented with the option of inducing labor for non-urgent indications versus waiting for spontaneous labor.

Patients with twin gestations may be offered induction of labor for several indications. The first is for medical or obstetric complications. Here, the indication for delivery is strong, and the alternative is usually an elective cesarean delivery. Another common indication for delivery in twins is gestational age. Despite the increased risk of spontaneous preterm birth in twins, almost half remain undelivered at 37 + 0 weeks [28], and due to evidence of increased perinatal morbidity and mortality after 38 weeks, ACOG recommends delivery for dichorionic twins between 38 + 0 and 38 + 6 weeks [11]. Lastly, induction of labor is often considered for non-medical indications, including maternal anxiety, discomfort, and the need for proximity to the hospital [29].

In the absence of a strong medical indication for induction, patients and their physicians must balance the potential advantages and disadvantages of induction of labor versus awaiting the spontaneous onset of labor. The results of our study suggest that whilst induction of labor in twin gestations is associated with high chances of success, there is an increased risk (12%) of unplanned intrapartum cesarean delivery. Although this increased risk did not translate into worse maternal and neonatal outcomes, patients should be made aware of this when considering induction of labor. 

The second aim of our study was to determine the safety and efficacy of induction of labor with oral PGE1 in twins and compare outcomes to other common methods of labor induction, namely IV Oxytocin and extra-amniotic balloon + IV Oxytocin. Induction with oral PGE1 has advantages in terms of ease of administration (oral or vaginal), obviates the discomfort of EAB balloon insertion, and offers convenient storage (no need for a refrigerator). Furthermore, a recent Cochrane review on the use of low-dose oral PGE1 for the induction of labor in singleton gestations demonstrated reduced rates of cesarean delivery compared to induction of labor with oxytocin, extra-amniotic balloon, and vaginal dinoprostone with no increase in the rates of non-reassuring fetal heart rate status or uterine hyperstimulation [30]. That said, like in many other developed countries, in Israel, the label indications for oral PGE1 do not include induction of labor [31]. Therefore, its administration in these settings requires institutional authorization by the Israeli Ministry of Health for off-label use according to rule 29c of the Israeli pharmaceutical guidelines [32]. Of note, its use for induction of labor is endorsed by the Israeli Society for Maternal and Fetal Medicine guidelines [33].

Indeed, much of the management of the induction of labor in twins is based on data from singleton gestations. However, induction of labor in this population may pose additional theoretical challenges related to uterine overdistention, including both uterine rupture and uterine resistance to oxytocin [26]. Small cohort studies investigating induction of labor with prostaglandins [21,34], extra-amniotic balloons [35], and oxytocin [26] in twins have shown these methods to be safe and effective but have conflicting results regarding their effect on cesarean delivery rates. Whilst some studies show that prostaglandins increase the risk of cesarean delivery compared to Oxytocin [21,34], others have demonstrated increased risks for cesarean delivery with an extra-amniotic balloon [36]. A recent secondary analysis of patients participating in the Twin Birth study compared cesarean delivery rates in twins undergoing induction of labor with prostaglandins (153 women, 42%) versus amniotomy +/− Oxytocin (215 women, 58%). The rate of cesarean delivery was 59.5% in both groups, and there were no differences in other maternal and neonatal outcomes [20].

We found that the method of induction of labor did not influence the risk of cesarean delivery nor create adverse maternal or neonatal outcomes. Furthermore, when analyzing the indications for cesarean delivery, non-reassuring fetal heart rate status and complications related to the second twin were the most common indications for cesarean delivery (Table 5), and only two cases (6%) were due to unsuccessful labor induction. Therefore, patients can be reassured that if they choose induction of labor, they have a high chance of entering active labor, and with appropriate patient selection, no one method is superior at achieving vaginal delivery. 

The risk of uterine rupture during induction of labor in twins was another important outcome we sought to investigate. Induction of labor increases the risk of uterine rupture in patients undergoing a trial of labor after cesarean delivery (TOLAC) [37]. Indeed, for this reason, misoprostol use is contraindicated in patients undergoing TOLAC [38]. We hypothesized that similar concerns might apply in twin gestations due to an overdistention of the uterus, particularly at term. Until now, only one study has investigated the risk of uterine rupture in twin induction of labor, and whilst in this study there were no cases of uterine rupture, the authors did not specify the method of induction of labor [19]. In our cohort, there were no cases of uterine rupture in patients undergoing induction of labor, including 142 patients receiving oral misoprostol. Due to the rarity of uterine rupture, this study was not powered to detect any significant differences in uterine rupture; however, the lack of uterine rupture in this cohort of over 268 twins undergoing induction of labor, including 142 patients with oral PGE1, is reassuring. 

### 4.1. Strengths 

This is the largest cohort to date investigating both maternal and neonatal outcomes in twins undergoing induction of labor (268 patients) and comparing outcomes to a control group of twins entering labor spontaneously. Using this population as our control group better demonstrates the contribution of induction of labor-to-labor outcomes in twin gestations and addresses a clinical dilemma in patients with twin gestations who are considering elective induction of labor at term. This is also the largest cohort in the literature to report outcomes from the use of prostaglandins for the induction of labor in twin pregnancies (142 patients) and only the second study investigating the impact of induction of labor in twins on uterine rupture. The use of PGE1 for the induction of labor is increasing due to its superiority both in terms of convenience and efficacy compared to other methods of induction. However, until now, there has been limited data regarding its use in twins and its effect on uterine rupture, and our results suggest that this is a safe and effective method of induction for twin gestations This study also compared outcomes of three distinct labor induction techniques, allowing patients and physicians to make an informed decision regarding the optimal mode of induction for twin pregnancies. 

### 4.2. Limitations

Aside from its retrospective nature, the current study is limited by a lack of information regarding chorionicity and medical co-morbidities. In addition, as previously stated, our study population comprised a large proportion of multiparous and grandmultiparous women who were highly motivated to achieve vaginal delivery. 

Although we believe that the low rates of complications are reassuring and applicable to other patient populations, this should be taken into consideration when counseling patients. Randomized prospective studies comparing methods of induction in twin deliveries are required to investigate these findings further.

## 5. Conclusions

The results of our study demonstrate that induction of labor in twin gestations is safe, feasible, and has a high chance of success. Although associated with an increased risk of cesarean delivery, it does not increase the risk of any other adverse maternal or neonatal outcomes, including uterine rupture. Furthermore, the method of induction of labor used, namely, oral PGE1, IV Oxytocin, or extra-amniotic balloon + IV Oxytocin, has no effect on success rates nor on adverse maternal or neonatal outcomes. This information can be utilized to counsel patients and aid in decision-making when contemplating the option of inducing labor in twin pregnancies.

## Figures and Tables

**Table 1 jcm-12-02041-t001:** Demographic characteristics of patients with twin gestations in the study group and control group (Induced vs. spontaneous labor).

	Induction of Labor *n* = 268	Spontaneous Labor *n* = 450	*p*-Value
Maternal age (years) *	29.7 ± 5.7	30.7 ± 5.6	0.03
Gestational age at delivery (weeks) *	36.9 ± 1.5	36.9 ± 1.5	0.75
Parity (n)	2.7 ± 2.5	3.2 ± 2.6	0.08
Nulliparas	64 (23.9%)	62 (13.8%)	<0.001
Multiparas (Parity 1–4)	150 (56%)	261 (58%)	0.59
Grandmultiparas (Parity >4)	54 (20.1%)	127 (28.2%)	0.01
Neonatal birth weight (n) *	2562 ± 402.0	2565 ± 390.4	0.93
Birthweight discordancy >20%	34 (12.7%)	41 (9.1%)	0.13
Vertex/Non-Vertex **	113 (42.2%)	199 (44.2%)	0.64

* Continuous variable; ** Vertex/Non-Vertex—First twin in vertex presentation and second twin in non-vertex presentation.

**Table 2 jcm-12-02041-t002:** Maternal and neonatal outcomes of patients with twin gestations in the study and control group (Induced vs. spontaneous labor).

	*n*	*p*-Value	Odds Ratio	95% CI
**Cesarean delivery**		0.03	1.7	1.04–2.85
Induction	33 (12.3%)			
Spontaneous	34 (7.5%)			
**Spontaneous vaginal delivery of both twins**		0.93	0.99	0.7–1.4
Induction	194(72.3%)			
Spontaneous	327 (72.7%)			
**Operative vaginal delivery of at least one twin**		0.16	0.74	0.5–1.1
Induction	41 (15.3%)			
Spontaneous	88 (19.6%)			
**Combined adverse outcome**		0.85	0.93	0.6–1.4
Induction	21 (7.8%)			
Spontaneous	39 (8.7%)			
**5 min APGAR <7**				
Induction	0	0.27	0.99	0.99–1.00
Spontaneous	2 (0.2%)			
**pH < 7.1 ***				
Induction	4 (1.5%)	0.86	1.12	0.31–4.0
Spontaneous	6 (1.3%)			
**Post-partum hemorrhage (PPH)**				
Induction	14 (5.2%)	0.37	0.75	0.39–1.42
Spontaneous	31 (6.9%)			
**Uterine rupture**				
Induction	0	-	-	-
Spontaneous	0	-	-	-

Combined adverse outcome (PPH, Uterine rupture, umbilical cord arterial pH <7.1, APGAR <7 at 5 min); * Umbilical cord artery pH.

**Table 3 jcm-12-02041-t003:** Demographic characteristics of patients with twin gestation undergoing different methods of labor induction.

	Oral PGE1 *n* = 142	Oxytocin ± AROM *n* = 96	EAB + Oxytocin *n* = 29	*p*-Value
Maternal age (years) *	30.0 ± 5.5	29.4 ± 5.6	29.9 ± 7.0	0.16
Gestational age at delivery (weeks) *	36.9 ± 1.4	36.9 ± 1.4	36.9 ± 1.4	0.84
Parity (n)	2.6 ± 2.5	2.7 ± 2.4	2.5 ± 2.8	0.08
Nulliparas	34 (23.9%)	19(19.8%)	10 (34.5%)	0.26
Multiparas (Parity 1–4)	77 (54.2%)	61 (63.5%)	12 (41.4%)	0.08
Grandmultiparas (Parity >4)	31 (21.8%)	16 (16.7%)	7 (24.1%)	0.53
Neonatal birth weight (n) *	2586± 398.0	2545± 418.4	2501± 375.5	0.71
Birthweight Discordancy >20%	18 (12.7%)	13(13.5%)	3 (10.3%)	0.45
Vertex/Non Vertex **	57(40.1%)	44 (45.8%)	12 (41.4%)	0.8

PGE1—prostaglandin E1; AROM—Artificial rupture of membranes; EAB—Extra-amniotic balloon; * Continuous variable; ** Vertex/Non-Vertex—First twin in vertex presentation and second twin in non-vertex. Presentation.

**Table 4 jcm-12-02041-t004:** Maternal and Neonatal outcomes of patients with twin gestation undergoing different methods of labor induction.

	N	PGE1 vs. Oxytocin	Oxytocin vs. EAB	PGE1 vs. EAB
	Odds ratio + 95% Confidence interval + *p*-value			
**Cesarean delivery**				
PGE1	19 (13.4%)	OR 1.1 (0.4,2.0) *p* = 1.0	OR 2.1 (0.1,2.4) *p* = 0.52	OR 2.1 (0.1,2.1) *p* = 0.53
Oxytocin	12(12.5%)			
EAB	2(6.9%)			
**Operative vaginal delivery**				
PGE1	18(12.7%)	OR 0.74 (0.7,2.9) *p* = 0.45	OR 0.6 (0.58,4.3) *p* = 0.4	OR 0.45 (0.8,5.9) *p* = 0.15
Oxytocin	16 (16.7%)			
EAB	7 (24.1%)			
**Uterine Rupture**				
PGE1	0	N/A	N/A	N/A
Oxytocin	0			
EAB	0			
**Combined adverse outcome**				
PGE1	10 (7%)	OR 0.77 (0.5–3.5) *p* = 0.63	OR 1.4 (0.15–3.5) *p* = 0.5	OR 0.98 (0.2–4.7) *p*= 0.54
Oxytocin	9 (9.4%)			
EAB	2 (6.9%)			

PGE1—Prostaglandin E1; EAB- Extra-amniotic balloon; Combined adverse outcome (PPH, Uterine rupture, umbilical cord arterial pH < 7.1, APGAR < 7 at 5 min).

**Table 5 jcm-12-02041-t005:** Indications for Cesarean delivery (CD) in twin gestations undergoing different methods of labor induction.

	PGE1 *n* = −142	Oxytocin± AROM *n* = 96	EAB + Oxytocin *n* = 29	All Methods of IOL (268)
Overall CD rate	19(13.4%)	12 (12.5%)	2 (6.9%)	12.3%
NRFHM	8 (5.6%)	2 (16.7%)	0	30.3%
Arrested labor	3 (15.8%)	3 (25%)	0	18%
Complications relating to second twin	4 (21%)	3 (25%)	2 (6.9%)	27.3%
Failed induction	0	2 (16.7%)	0	6%
Other	4 (21%)	2 (16.7%)	0	18.1%

PGE1—Prostaglandin E1; AROM—Artificial rupture of membranes; EAB—Extra-amniotic balloon; IOL—Induction of labor; CD—Cesarean delivery; NRFHM—Non-reassuring fetal heart monitor.

**Table 6 jcm-12-02041-t006:** Studies investigating outcomes of induction of labor in twin gestations.

Study	N	Control	Success Rate	Odds Ratio for CS	Uterine Rupture	Maternal Outcome	Neonatal Outcome
Loscul 2019 [17]	1995	Singleton IOL	77%	1.8	N/A	N/A	N/A
Taylor 2012 [18]	100	Singleton IOL	81%	1.7	N/A	N/A	N/A
Okby 2013 [19]	191	Singleton IOL	69%	2.2	Not increased	Not increased	N/A
Mei-Dan 2017 [20]	368	PGE1 vs. Oxytocin in twins	59.5%	1	N/A	No difference	No difference
Jonsson 2015 [21]	220	Spontaneous labor in twins	79%	1.9	N/A	N/A	No difference
Mikaelsen 2022 [22]	63	Spontaneous twin labor	72.5%	0.65	N/A	No difference	No difference
Our data	268	Spontaneous twin labor	87.7%	1.7	0%	No difference	No difference

IOL—Induction of labor; PGE1—prostaglandin E1; N/A—Not available.

## Data Availability

Not applicable.

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
