# Peer review of "Induction of Labor in Twins—Double Trouble?"

_jcm, 2023, doi:10.3390/jcm12052041_

Round 1
Reviewer 1 Report
Dear authors, this is an elegant study on 3 different methods of labor induction in twin pregnancies in patients being highly motivated for vaginal births. You have presented very useful data and results with practical conclusions and suggestions on how to induce twins in our clinical practice. The big insufficiency you mentioned is lack of data on chorionicity and this should be your next aim in the future study. So far you may want to apply the following remarks to improve the manuscript.
1/ Confusion between line 115-116 and p value in table 1.
2/ Suggestion to change in Table 1. Demographic characteristics of patients with twin gestations in study vs control group (induced vs spontaneous labor) 3/ Suggestion to change in Table 2. Maternal and neonatal outcomes of patients with twin gestations in study vs control group (induced vs spontaneous labor) 4/ Umbilical cord arterial blood pH is not the same as umbilical cord artery pH. Change for the latter in table 2 5/ Suggestion to change in Table 3. Demographic characteristics of patients with a twin gestation undergoing different methods of labor induction. 6/ Suggestion to change in Table 4. Maternal and Neonatal outcomes of patients with a twin gestation undergoing different methods of labor induction. 7/ Suggestion to change in Table 5. Indications for Cesarean delivery (CD) in twin gestations undergoing different methods of labor induction. 8/ Line 173: should be table 6 9/ In table 6 suggestion to change Lopian 2023 onto Our data 10/ Expand all abbreviations in tables below the tables. 11/ Is use of oral PGE1 for induction of labor legally approved or applicated off-label in your country? Make a short comment on this issue in your discussion and the situation in other countries.
Author Response
Dear Reviewer
Thank you for taking the time to review this paper and for your constructive remarks.
Regarding point 1, this was a typing error and has been amended in the text.
Regarding point 2.
All ammendments have been made.
The use of PGE1 is off-label in Israel and this has been refered to in our discussion
Many Thanks
Miriam Lopian
Reviewer 2 Report
I appreciate the opportunity to review the manuscript entitled “Induction of labor in twins. Double trouble?” submitted to the Journal of Clinical Medicine.
The authors settled the retrospective observational at a single center with a high-volume delivery ward (11000 deliveries per year) in order to investigate safety and efficacy of different methods of induction of labor in twin pregnancies. The control group consisted of patients in tween gestations who went in the labor spontaneously. A total of 268 women were included in the study group and 450 in the control group respectively. The primary outcome was cesarean delivery of either one or both tweens. Secondary outcomes included mode of delivery and maternal and neonatal complications.
The introduction section contains necessary and relevant data on the subject available in the literature.
The methods section provides concise and clear explanation of the study protocol, with all the data needed for adequate explanation of inclusion and exclusion criteria. The authors mentioned that the local ethical review board approved the study. Please add the number of the ethical approval.
Results are described in detail both in text and tables. In this section it would be nice to provide the data in the text on “complications relating to second twin” for the nine cases in the study. I would suggest the authors describe these complications.
Discussion sections address all the obtained results in relation to relevant literature data. The limitations and strengths of the study are appropriately described.
References are not cited uniformly and must be revised (i.e. reference 21 contains “2016 Jun 16). Please, check reference #8.
Taking into account that there is a lack of studies about outcomes of inductions of deliveries of twin pregnancies and studies which have spontaneous delivery of twin pregnancies as control group, this study contributes significantly to new clinical knowledge which could be the start points for the guidelines for the induction of delivery and new randomized clinical trials.
Author Response
Dear Reviewer,
Thank you for taking the time to review this manuscript and for your constructive remarks.
The number of the ethical approval has been added to the manuscript.
Details of complications requiring cesarean delivery of the second twin have been added in text to the results section.
Ammendments have been made to the references
Many Thanks
Miriam Lopian